🔬 PLOS | ONE

# Muscle activity of Bulgarian squat. Effects of additional vibration, suspension and unstable surface

Joan Aguilera-Castells[1]©, Bernat Buscà[1]©*, Jose Morales[1]‡, Mònica Solana-Tramunt[1]‡, Azahara Fort-Vanmeerhaeghe[1,2]‡, Fernando Rey-Abella[2]‡, Jaume Bantulà[1]‡, Javier Peña[3]©

1 Faculty of Psychology, Education Sciences and Sport Blanquerna, Ramon Llull University, Barcelona, Spain, 2 School of Health Science Blanquerna, Ramon Llull University, Barcelona, Spain, 3 Sport and Physical Activity Studies Center (CEEAF), University of Vic – Central University of Catalonia, Barcelona, Spain

© These authors contributed equally to this work.
‡ These authors also contributed equally to this work.
* bernatbs@blanquerna.url.edu

**Data Availability Statement:** All relevant data files are available from the FIGSHARE database here: https://figshare.com/s/53fc813b0039ba7552af or

## Abstract

Practitioners of strength and conditioning are increasingly using vibration and unstable environments to enhance training effects. However, little evidence has been found comparing the use of suspension devices and vibratory platforms used in the Bulgarian squat. The purpose of this cross-sectional study was to examine the effect of suspension devices (TRX®), unstable surfaces (BOSU®), and vibration plates on muscle activity and force during the Bulgarian squat. Twenty physically active male students (age = 24.40 ± 3.63 years) performed a set of five repetitions of Bulgarian squats, suspended lunges, suspended lunges-BOSU, suspended lunges-Vibro30, and suspended lunges-Vibro40 (vibration 30 Hz or 40 Hz and 4 mm of amplitude). A randomized within-subject design was used to compare leg muscle activity, vertical ground reaction forces, and force exerted on the strap across the five exercises. Results showed no significant differences in muscle activity between the Bulgarian squat and suspended lunge ($p = 0.109$, $d = 2.84$). However, the suspended lunge significantly decreased muscle activation compared to the suspended lunge-BOSU ($p = 0.012$, $d = 0.47$), suspended lunge-Vibro30 ($p = 0.001$, $d = 1.26$), and suspended lunge-Vibro40 ($p = 0.000$, $d = 1.51$). Likewise, the Bulgarian squat achieved lower activity than the suspended lunge-Vibro40 ($p = 0.010$, $d = 0.96$). The force on the strap significantly decreased in the suspended lunge-BOSU compared to the suspended lunge-Vibro30 ($p = 0.009$, $d = 0.56$). The suspended lunge achieved higher front leg force production than the Bulgarian squat ($p = 0.006$, $d = 0.48$). In conclusion, leaning the rear leg on a suspension device does not provoke an increase in the activation of the front leg during the Bulgarian squat but increases the vertical ground reaction forces. Thus, the use of unstable surfaces or vibration plates for the front leg increased muscular activity when performing a suspended lunge.

here: https://doi.org/10.6084/m9.figshare.
8174639.

**Funding:** Author JAC received funding for
conducted this study from the Secretariat of
University and Research of the Ministry of
Business and Knowledge of the Government of
Catalonia and the European Social fund under
Grant [2019 FI_B1 00165]; and the Ministry of
Education, Culture and Sport of the Government of
Spain under Grant [311327]. The funders had no
role in study design, data collection and analysis,
decision to publish, or preparation of the
manuscript.

**Competing interests:** The authors have declared
that no competing interests exist.

## Introduction

In strength and conditioning, recent trends support the use of functional exercises to improve
the efficacy of multidirectional sports skills, enhancing the quality of resistance training. These
skills include locomotor, manipulative, and stability actions while maintaining control of the
kinetic chain [1]. Most of these actions involve unilateral actions of multidirectional jumping,
change of direction, and sprinting using different techniques, with a significant anteroposter-
ior, lateral rotational force-vector application [2–4]. Thus, Bulgarian [5] and single-leg squats
[6] or side-steps and backward lunges [2] have been a part of effective sport-specific training
programs. Nuñez et al. [7] found significant improvements in a 90˚ change of direction in a
unilateral resistance training group compared that in a bilateral training group in team sports.
Moreover, Bogdanis et al. [8] showed some evidence supporting the benefits of unilateral resis-
tance training in jumping and rate of force development in physical education students. In the
same direction, Gonzalo-Skok et al. [2] demonstrated higher improvements in functional tests
(180˚ change of direction, lateral jump, and one-leg horizontal jump) for a unilateral resistance
training group in team sports. The same leading author also found a between limb imbalance
reduction following this training paradigm in basketball players [9]. Therefore, due to their
specificity and transferability to sports skills, the step-up, standard lunge (two feet on the
floor), or Bulgarian squat (rear foot elevated) are among the most widely used exercises to
enhance lower body strength [10].

### Instability

Coaches, athletes, and fitness enthusiasts are continuously searching for new challenges to
increase training demands through the complexity of the exercises, for instance, by modifying
the amount of instability or intensity [11]. Thus, the use of devices that create instability has
become popular (i.e., BOSU® Ball, Wobble Board®). Primarily, unstable devices are used to
increase the load of traditional exercises by providing higher muscular demands through supe-
rior motor unit recruitment. Such devices also improve neuromuscular coordination to main-
tain balance during training exercises [12]. As Behm et al. [11] stated, strength training on
unstable surfaces or unstable implements provides an augmented degree of instability com-
pared to stable surfaces. Hence, destabilizing environments provide more varied and effective
training stimuli, enhancing neuromuscular adaptations [13]. Likewise, some evidence sup-
ports the idea that instability training elicits higher activity of several upper body and trunk
muscles than traditional exercises such as push-ups, sit-ups, and back extensions. Anderson
et al. [14] recruited highly trained individuals to examine triceps brachii, erector spinae, rectus
abdominis, internal oblique and soleus activation while performing traditional and unstable
push-ups in the single (hands or feet on the unstable surface) or dual (both hands and feet
on the unstable surface) condition. The authors found that the dual condition provoked the
highest percentage of change (>150%) for all the analyzed muscles than the other conditions.
Besides, a significant linear effect was found between the amount of instability provided and
level of muscle activity in all muscles and exercise conditions. Cosio-Lima et al. 's study [15]
showed that after 5 weeks of sit-up and back extension unstable training (Swiss ball) in
untrained college women, muscle activity of rectus abdominis and erector spinae significantly
increased compared to that of a control group. Furthermore, some evidence of this has been
found in lower body exercises such as standard lunges [16] and Bulgarian squats [17]. Con-
cretely, performing standard lunges and Bulgarian squats involves the activation of the gluteus
maximus and medius, vastus medialis, vastus lateralis, rectus femoris, biceps femoris, semiten-
dinosus, and gastrocnemius [18,19]. In order to assess muscle activity during a standard lunge,
Boudreau et al. [18] used surface electromyography to measure the activity of rectus femoris,

gluteus medius, and gluteus maximus in healthy individuals and demonstrated that the activation of gluteus medius, gluteus maximus, and rectus femoris ranged from low to moderate (from <21% to 40%) maximum voluntary isometric contraction (MVIC). Others authors [19] have examined the effect of performing a Bulgarian squat (loaded) on the activity of gluteus maximus, biceps femoris, semitendinosus, rectus femoris, vastus lateralis, vastus medialis, and gastrocnemius and reported that Bulgarian squats provoked higher muscular recruitment (>638 mV) in the quadriceps muscles (rectus femoris, vastus medialis, and lateralis) than in the hamstrings (biceps femoris and semitendinosus), gluteus maximus, and gastrocnemius (all of them <396 mV). DeForest et al. [19] reported that all analyzed muscles achieved higher activation during the concentric phase than in the eccentric phase.

Regarding the effects of unstable conditions in the lower body, only Andersen et al. [17] examined the effect of performing a standardized Bulgarian squat (6-RM loaded) under stable (front leg on the floor) and unstable (front leg on a foam cushion) conditions on the hip and thigh muscles of healthy trained participants. Bulgarian squats significantly increased the activation of biceps femoris under stable conditions compared to those under unstable conditions (stable vs. unstable: 215.5 ± 106.7% MVIC vs. 193.3 101.5% MVIC, $p = 0.030$), and there were no significant differences for rectus femoris, vastus medialis, vastus lateralis, and gastrocnemius, and all of them achieved a high activation (>60% MVIC) under both exercise conditions. In contrast, Youdas et al. [16] found that surface (stable vs. unstable) and sex have a significant effect on the activations of rectus femoris (women vs. men in stable surface: 33.9% MVIC vs. 20.1% MVIC, respectively; $p = 0.04$) and hamstring (men vs. women in unstable surface: 37.9% MVIC vs. 19.9% MVIC, respectively; $p = 0.04$) during the extension of a standard lunge in healthy recreational athletes. Thus, evidence that the use of unstable surfaces increases muscular demands during Bulgarian squat and standard lunge exercises is weak.

## Whole-body vibrations

Other devices such as whole-body vibration (WBV) platforms are commonly used to increase neuromuscular performance in strength training. These platforms modify workloads through vibration (side-alternating vibration or synchronous vibration), frequency (in Hz), and amplitude (peak to peak displacement, in mm) and, as a consequence, the magnitude of acceleration following the muscle tuning paradigm [20,21]. WBV is applied to the muscle or tendon to elicit tonic vibration reflex [22], and the beneficial effects of WBV have been demonstrated in lower limb exercises (squat, half-squat, Bulgarian squat, or lunge) in different cohorts such as untrained, recreationally active, and older adults [23,24]. As for muscle activation, vastus lateralis recruitment increases when performing 60 s of static half-squat with 100˚ of knee flexion at three different WBV frequencies (30, 40, and 50 Hz) with 10 mm of amplitude [25]. Likewise, Di Giminiani [26] reported that performing 20 s of static half-squat in four different positions (knee flexion angle ranging from 90˚ to 120˚) with WBV (45–55 Hz and 1 mm of amplitude) increased the activation of vastus lateralis compared to a half-squat with no vibration applied in male sport sciences students. Moreover, Ritzmann et al. [27] found that a progressive increase in WBV frequencies (from 5 to 30 Hz) and amplitudes (from 2 to 4 mm) causes a progressive increase in the activation of vastus medialis, rectus femoris, and biceps femoris while performing 10 s of static half-squat. Thus, frequencies ranging from 30 to 55 Hz and amplitudes from 2 to 5 mm elicited the highest response in the muscles mentioned above [23,27,28]. Although WBV increases the activation of thigh muscles during lower body exercises, such as the squat, Bulgarian squat, or lunge, there is a rising interest in enhancing muscular activity through the use of different suspension devices. Furthermore, the use of a combination of different methods to increase muscular activation has been investigated

[29–31]. Vibratory platforms, flywheels, rubber bands, or pulley machines have been used together with other devices such as Pielaster®, Swiss Balls, Freeman plates, and BOSU® to create instability. Moras et al. [32] recently compared the variability in force production of a stable and unstable bilateral squat using a flywheel machine and found no significant differences between both conditions in terms of sample entropy in healthy trained men. Nevertheless, combinations of suspension devices with other training methods are still unexplored.

## Suspension devices

In suspension training, a suspension device is required to create an unstable condition. This method utilizes a system of straps with handles on the bottom and attached to a single anchor point [33]. This device acts as a pendulum by rotating around the singular anchor point. The suspension device uses body weight and fundamental principles (vector resistance, stability, and pendulum) to enhance motor unit recruitment [34]. The effects of using a suspension device on lower body muscle activity have been investigated while performing a hamstring curl. Specifically, Malliaropoulos et al. [35] examined the effect of ten hamstring loading exercises (standard lunge, single-leg Romanian deadlift T-drop, kettlebell swing, bridge, suspended hamstring curl, hamstring bridge, curl, Nordic exercise, Swiss ball flexion and slide leg exercise) on biceps femoris and semitendinosus recruitment in elite female track and field athletes and reported that the biceps femoris and semitendinosus achieved a very high activation (>60% MVIC) in the suspended hamstring curls compared to the high-to-low activity (<60% MVIC) for the standard lunge, single-leg Romanian deadlift T-drop, kettlebell swing, bridge, hamstring bridge, curl, and Nordic exercise. However, the suspended hamstring curl was less demanding for the biceps femoris (84% MVIC) and semitendinosus, (75% MVIC) than the Swiss ball flexion and the slide leg exercise, both with muscle activity >90% MVIC. Recently, Krause et al. [36] assessed the activation of hip and thigh muscles during a suspended lunge (rear leg leaning on the suspension device cradles) and its counterpart. The suspended lunge exercise achieved significantly higher activation in the hamstring, gluteus maximus, gluteus medius, and adductor longus than the standard lunge. Despite this, the authors did not find significant differences in the rectus femoris between the exercise conditions.

## Forces in suspension training

Apart from muscular activation, force production is also useful in assessing the load involved in strength exercises. Several studies have examined the forces exerted in different lower limb exercises. Comfort et al. [37] reported that single-leg squat achieved greater peak vertical ground reaction forces (VGRF) and higher ankle-joint moment, but a lower hip-moment, compared to the joint kinetics and kinematics analyses of forward and reverse lunges. Other studies have assessed the load on the suspension strap and VGRF in upper body exercises. Melrose and Dawes [38] measured the force exerted on the suspension strap while performing an isometric suspended inverted row in college students. These authors found that the percentage of body mass resistance on the suspension strap increases from 37.4% to 79.4% when the trunk-leg inclination is closer to the floor (from 30˚ to 75˚). Likewise, Gulmez [39] recruited male sport sciences students to examine the force on the suspension strap and VGRF while performing isometric suspended push-ups under two conditions (elbow flexion and elbow extension). The study found that when trunk-leg inclination is modified (from 45˚ to 0˚), the percentage of body mass resistance increases (elbow flexion: 36.8% to 75.3%; elbow extension: 11.9% to 50.4%), while VGRF decreases (elbow flexion: 80.7% to 32.2%; elbow extension: 97.5% to 46.6%). However, the effect of load on the suspension strap while performing lower

body exercises such as squats, standard lunges, Bulgarian squats, or hamstring curls has apparently not been assessed yet. Conversely, the effects of other sources of instability on force production have been examined for lower body exercises. Previous studies have shown that an unstable environment leads to decreased force output [40,41]. Saeterbakken & Fimland [42] examined squat exercise on four different unstable surfaces and the BOSU® condition, obtaining the lowest force output value compared to a stable squat condition. Likewise, another investigation reported that BOSU® and T-Bow® deadlift conditions significantly decreased force production in deadlift on the floor [43]. Although the literature review suggests that unstable surfaces reduce force production, the dual condition (two destabilizing materials or WBV with an unstable surface) might increase muscle activation [29,44]. However, Byrne et al. [45] reported no significant difference when studying the dual condition on the suspended plank.

To the best of our knowledge, there is insufficient evidence of muscle activity and force production when a suspended lower body exercise is performed. Therefore, our primary purpose is to study the effect of suspension devices on muscle activity during a Bulgarian squat. Second, we aim to determine the effect of adding an unstable surface and WBV on muscle activity in the suspended lunge. Regarding force production, the objective was to quantify the effect of adding an unstable surface and WBV on the forces exerted on the suspension strap by the rear leg. We also compared the VGRF produced by the front leg between the Bulgarian squat and suspended lunge. Therefore, it was hypothesized that 1) a suspended lunge results in greater muscle activation than a Bulgarian squat, 2) muscle activation under Bulgarian squat and suspended lunge conditions (suspended, suspended-BOSU, suspended-vibration 30 Hz, and suspended-vibration 40 Hz) significantly differs in all analyzed muscles (rectus femoris, biceps femoris, gluteus medius, vastus lateralis, vastus medialis, and rectus femoris of the rear leg), 3) the force exerted on the suspension strap is significantly lower in suspended lunge-BOSU than under the other suspended lunge conditions, and 4) the suspended lunge condition elicits a higher VGRF load on the front leg than the Bulgarian squat.

## Materials and methods

### Design

A repeated measures design was used to compare electromyographic activity and force output (force exerted on the suspension strap and VGRF) during the Bulgarian squat and under four suspended lunge conditions. Twenty participants were recruited to perform the Bulgarian squat and suspended lunges. Bulgarian squats were performed with the front foot on the floor and the rear foot leaning on a bench. Suspended lunge conditions were a) suspended lunge (front foot on the floor and the rear foot leaning within the suspension device cradle), b) suspended lunge-BOSU (same as the previous exercise with front foot on BOSU®), c) suspended lunge-Vibro30 (front foot on the WBV platform at 30 Hz and 4 mm of amplitude), and d) suspended lunge-Vibro40 (same as the previous exercise with 40 Hz and 4 mm of amplitude). All suspended lunge conditions were executed using a TRX Suspension Trainer™ device. An S-Type Load Cell was used to measure the force exerted on the suspension strap by the suspended lower limb. The load cell was displayed on the suspension device. A force plate was utilized to register VGRF from the front leg in both the Bulgarian squat and suspended lunge. Surface electromyography (sEMG) was used to measure muscle activity in the dominant leg (front leg). The following muscles were analyzed: 1) rectus femoris, 2) biceps femoris, 3) gluteus medius, 4) vastus medialis, and 5) vastus lateralis. Additionally, activity in the rectus femoris of the rear leg was registered across the five exercises.

## Participants

Twenty healthy and physically active male university students (mean age = 24.40 ± 3.63 years, range: 20–31 years, height = 1.79 ± 0.06 m, body mass = 78.06 ± 1.70 kg, body mass index = 24.35 ± 1.58 kg·m$^{-2}$) were voluntarily recruited for this study. Subjects had been physically active with at least three sessions per week with a minimum duration of 30 min. Additionally, eight of the included subjects played soccer, six played basketball, three played handball, and three played tennis. Subjects were excluded if they presented any injuries and/or pain related to cardiovascular, musculoskeletal, or neurological disorders. All subjects were asked to come to the experimental session after refraining from high intensity physical activity for 24 h before the testing, and they consumed no food, drinks, or stimulants (i.e., caffeine) 3–4 h before testing. During the familiarization session, all subjects signed the written informed consent after receiving a clear explanation of the experimental procedures, exercise protocol, benefits, and possible risks associated with their participation. The Ethics and Research Committee Board at Blanquerna Faculty of Psychology and Educational and Sports Sciences of Ramon Llull University of Barcelona approved this study with reference number 1819005D. All protocols conducted in this research complied with the requirements specified in the Declaration of Helsinki (revised in Fortaleza, Brazil, 2013). In accordance with the PLOS consent guidelines, participants gave their written informed consent for their images to be reproduced in this manuscript.

## Procedures

The study was conducted in two sessions: familiarization and experimental. They were performed at the same time in the morning, separated by a week. During the familiarization session, researchers recorded the age, weight, and height of each subject, and measured leg length, which was defined as the distance from the anterior superior iliac spine to the medial malleolus of the tibia [18]. Leg dominance was determined by asking subjects which leg they would use to kick a ball [46]. The dominant leg was used as the front leg in the Bulgarian squat and under suspended lunge conditions. To verify adherence to pre-test instructions, all subjects completed a standardized questionnaire. Subjects were familiarized with the exercise procedures by performing two sets of five repetitions under each exercise condition (Bulgarian squat, suspended lunge, suspended lunge-BOSU, suspended lunge-Vibro30, and suspended lunge-Vibro40), to achieve proper technique before data collection. A 1-min resting period between repetitions and a 2-min resting period between exercises were allowed to avoid fatigue.

During the experimental session, subjects were outfitted with surface electrodes and completed a MVIC test. Before the MVIC test, subjects performed a standardized warm-up, which consisted of 5 minutes of cycling with 100 W of cadence maintaining 60 revolutions per minute. After the MVIC test protocol, each subject performed a set of five consecutive repetitions of the Bulgarian squat and the suspended lunge exercises. The objective was to perform the different tasks at a controlled pace, maintaining posture as consistently as possible. The suspended lunge was performed under 4 conditions: 1) suspended lunge, 2) suspended lunge-BOSU, 3) suspended lunge-Vibro30 (WBV at 30 Hz and 4 mm of amplitude), and 4) suspended lunge-Vibro40 (WBV at 40 Hz and 4 mm of amplitude). In the suspended lunge-Vibro30 and -Vibro40, the WBV plate was set at 30 and 40 Hz, respectively. These frequencies show the highest demands for the knee thigh muscles in similar tasks [23,27,28]. The strength and conditioning methods used in the study procedures, including suspension, unstable surfaces, and WBV, are frequently used in several sports where the inclusion of additional weight is less common (i.e., soccer, field hockey, tennis, paddle tennis, and badminton).

The Bulgarian squat and suspended lunge exercise orders were randomized between subjects and 90 seconds of rest between exercises was allowed to prevent fatigue. Pace was standardized using a metronome (*Pro Metronome* application, version 3.13.2; EUM Lab-Xannin Technology Gmbh., Hangzhou, CHN) set at 70 beats per minute (bpm), and the tether of a positional encoder (WSB 16k-200; ASM Inc., Moosinning, DEU) was attached to the hip and used to measure its vertical displacement during all exercises. Trials were discarded and repeated if subjects were unable to perform the exercises with the correct technique.

## Surface electromyography signal

All sEMG values were recorded using a BIOPAC MP-150 at a sampling rate of 1.0 kHz. Data were analyzed using the AcqKnowledge 4.2 software (BIOPAC System, INC., Goleta, CA). sEMG signals were bandpass filtered at 50–500 Hz while utilizing a 4th order Butterworth filter. Root mean square sEMG signals were recorded throughout each exercise. The mean root mean square data were then normalized to the maximal voluntary isometric contraction and reported as % MVIC.

Bipolar sEMG electrodes (Biopac EL504 disposable Ag-AgCl) with an inter-electrode distance of 2 cm were used. Surface electrodes were placed on the dominant leg (front leg) on the rectus femoris, biceps femoris, gluteus medius, vastus medialis, and vastus lateralis. An additional electrode was placed on the rectus femoris of the rear leg. Before affixing the electrodes, the subject's skin sites were prepared for application through shaving, exfoliation, and alcohol cleansing in order to reduce impedance from dead surface tissue and oils [47]. After that, the electrodes were placed following the SENIAM Project recommendations [47]. Electrodes for the rectus femoris (front and rear leg) were placed at 50% on the line running from the anterior spine iliac superior to the superior part of the patella, those for the biceps femoris were placed at 50% on the line between the ischial tuberosity and lateral epicondyle of the tibia, those for the gluteus medius were placed at 50% on the line from the crista iliac to the trochanter, those for the vastus medialis were placed at 80% on the line between the anterior spine iliac superior and joint space in front of the anterior border of the medial ligament, and those for the vastus lateralis were placed at 2/3 on the line from the anterior superior spine iliac to the lateral side of the patella. A ground surface electrode was placed directly over the right anterior superior iliac spine.

## Force measurements

VGRF was measured using a force plate (Kistler 9260AA, Winterthur, Switzerland) equipped with a data acquisition system (Kistler 5695b, Winterthur, Switzerland). Raw data were acquired (sampling rate 1,000 Hz) using the MARS software (Kistler, Winterthur, Switzerland). Calibration of the system was performed according to the MARS software recommendations. While the Bulgarian squat and suspended lunge were performed, subjects centered their forward foot at a fixed position on the force plate.

To record the load on the suspension device, an S-Type Load Cell (model CZL301C; Phidgets Inc., Alberta, CAN) was displayed between the anchor point (2.95 m from the ground) and suspension device straps. Data were collected (sampling rate 200 Hz) using BIOPAC MP-150 (BIOPAC System, INC., Goleta, CA) and its original software (AcqKnowledge 4.2; BIOPAC System, INC., Goleta, CA). The system was calibrated according to the manufacturer's recommendations in the manual.

## Maximum voluntary isometric contraction (MVIC)

Prior to the exercise trials described below, subjects performed three 5-s MVICs for each muscle, and the trial with the higher sEMG signal was selected in accordance with Jakobsen et al.

[48]. Subjects were instructed to increase muscle contraction force gradually towards maximum for a period longer than 2 s, sustain the MVIC for 3 s, and release the force again slowly. Three minutes of rest was allowed between each MVIC, and standardized verbal encouragement was provided to motivate all subjects to achieve maximal muscle activation. Positions during the MVICs were based on the Konrad [49] protocol for the dominant leg (front leg) muscles: rectus femoris, vastus medialis, vastus lateralis, biceps femoris, gluteus medius; and for rectus femoris of the rear leg. To obtain the MVIC of the rectus femoris, vastus medialis, and vastus lateralis, subjects performed an isometric 90˚ single leg knee extension in a seated position against matched resistance (i.e., resistance forceful enough to elicit an isometric contraction from the subject). The resistance was matched using an ankle bracelet attached to a cable that was anchored to a stretcher, thereby guaranteeing a fixed position. To obtain the MVIC of the biceps femoris, subjects performed an isometric 20–30˚ single-knee flexion in a prone-lying position against a matched resistance. Lastly, the MVIC for the gluteus medius was performed with subjects in a fixed side-lying position. An isometric hip abduction was then performed against a matched resistance. The exercise trials were performed once all MVICs were collected.

## Exercise trials

To normalize the height and stepped distance under all the Bulgarian squat and suspended lunge conditions, the height of both the Bulgarian squat bench and suspension device straps was normalized to 60% of the subject's leg length; this length added the height of the force plate, BOSU®, and WBV platform (i.e., total height strap = 60% of subject's leg length + BOSU®'s height). The distance that the subjects stepped in all the Bulgarian squat and suspended lunge conditions was normalized to 80% of their leg length, measured as the distance from the anterior superior iliac spine to the medial malleolus of the tibia, in accordance with Boudreau et al. [18]. Regarding the exercise load, all subjects used their bodyweight as a load in the Bulgarian squat and under the suspended lunge conditions. The proper techniques for the exercises were as follows:

- Bulgarian squat: Subjects were instructed to stand upright with one foot in front and the other behind the body. Subjects held their arms crossed on their chest, and their upper body was maintained upright with a lower back natural sway throughout the exercise. Subjects lowered the body (eccentric phase) until the forward knee flexed to 90˚, and subsequently returned the body to the starting position with a full knee extension of the forward leg (concentric phase), maintaining an erect trunk position, as required for subjects. The forward foot was placed at a fixed position with the heel contact on a force plate. The rear foot (instep) was leaned on a horizontal press bench. To adjust the height of the rear leg, EVA foam play mat pieces were used and fixed with a cinch strap (Fig 1). The contact point between the horizontal press bench and foot was controlled so that it was identical in all repetitions.

- Suspended lunge: Prior to performing this exercise, a TRX Suspension Trainer (Fitness Anywhere, San Francisco, CA) was secured in the anchor point. Subjects were instructed to assume a lunge position with the rear foot placed within the suspension device cradle with a slight plantar flexion (Fig 1). The forward foot was placed on a force plate. Then, subjects performed the lunge as previously described.

- Suspended lunge-BOSU: A BOSU® ball (BOSU®, Ashland, OH) was used to perform this exercise. Subjects assumed the above-stated position but with the forward foot placed upon the BOSU®, dome side up (Fig 1).

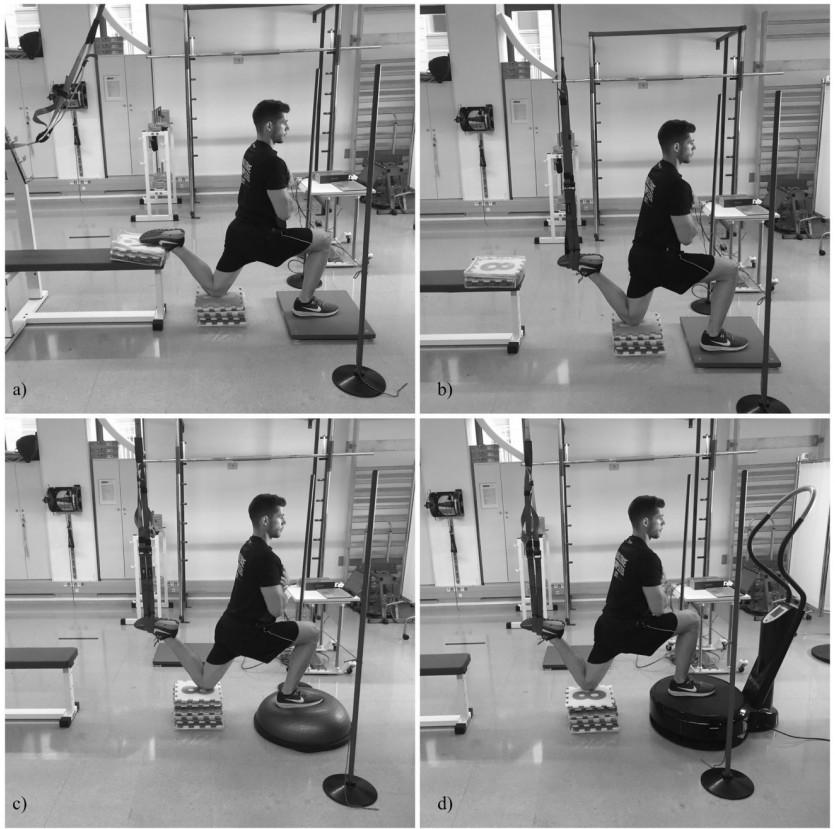

**Fig 1. Bulgarian squat (a), suspended lunge (b), suspended lunge-BOSU (c), and suspended lunge-Vibro30 and Vibro-40 (d).**

- Suspended lunge-Vibro30: A WBV platform (Compex® Winplate; DJO UK Ltd., Guildford, GBR) was used to perform this exercise. Subjects were instructed to place the forward foot and maintain the heel in contact upon the Compex Winplate. The WBV platform setting was 30 Hz of frequency and 4 mm of amplitude (high) (Fig 1). Subjects then performed the lunge as previously described.

- Suspended lunge-Vibro40: Subjects performed the lunge with a WBV platform set at 40 Hz of frequency and 4 mm of amplitude (high). They placed the rear foot in the suspension straps using the same techniques as previously described (Fig 1).

## Data analysis

All sEMG signal analyses were performed using the AcqKnowledge 4.2 (BIOPAC System, INC., Goleta, CA). The sEMG signals related to isometric exercises were analyzed by using the three middle seconds of the 5-s isometric contraction. The sEMG signals of the Bulgarian squat and suspended lunge conditions were analyzed by taking the average of the three middle repetitions. The first and fifth repetitions were excluded from data analysis. The sEMG amplitude in the domain was quantified using the root mean square. The mean root mean square values were selected for every trail and normalized to the maximum EMG (%MVIC). The global mean of all muscles (i.e., rectus femoris, biceps femoris, gluteus medius, vastus medialis, vastus lateralis, and rectus femoris of the rear leg) was also calculated (arithmetic mean) and

analyzed. To facilitate comparison of muscle activation between conditions, activation was categorized into four levels: >60%, very high; 41–60%, high; 21–40%, moderate; and <21%, low [50].

Recorded load data from the force plate and load cell were analyzed using the entire lunge phase (eccentric-concentric repetition). Maximum force values reached in the entire phase were used during the Bulgarian squat and suspended lunge conditions. The first and fifth repetitions were excluded from data analysis.

To normalize the load, an equation was calculated for each subject based on load and body weight (load_norm = load / body weight x 100) in accordance with Gulmez [39]. The normalized values were expressed as a percentage of the total load.

## Statistical analysis

Statistical analysis was accomplished using SPSS (Version 20 for Mac; SPSS Inc., Chicago, IL, USA). The sEMG signal of each muscle analyzed through all the Bulgarian and suspended lunge conditions, forces exerted on the suspension strap, VGRF, and MVICs assessment were measured. The intra-rater reliability of all the dependent variables was assessed using an intra-class correlation coefficient (ICC), and their 95% confidence intervals based on mean-rating (k = 3), absolute-agreement, two-way mixed effects model. The ICC was interpreted using the recommendations of Koo & Li [51] such as poor (<0.5), moderate (0.5–0.75), good (0.75–0.90), or excellent (>0.90) reliability. The number of subjects chosen was based on effect size 0.30 SD with an $\alpha$ level of 0.05 and power at 0.95 using G Power Software (University of Dusseldorf, Germany). The Shapiro-Wilk test was used to confirm that data were normally distributed to approve the use of parametric techniques. The results are reported as mean ± standard deviation. One-way repeated-measures analysis of variance (ANOVA) was employed to examine the effect of exercise condition on mean muscle activation and the forces exerted on the suspension straps. A paired t-test was conducted to compare VGRF produced by the front leg on the force plate in Bulgarian squat and suspended lunge. The Greenhouse-Greisser correction was used when the assumption of sphericity (Mauchly's test) was violated. Post hoc analysis with Bonferroni correction was used in case of significant main effects. Effect sizes are reported as partial eta-squared ($\eta_p^2$), with cut-off values of 0.01–0.05, 0.06–0.13, and >0.14 for small, medium, and large effects, respectively. For pairwise comparison, the Cohen's $d$ effect size was calculated [52], and the magnitude of the effect size was interpreted as <0.2 = trivial; 0.2–0.6 = small; 0.6–1.2 = moderate; 1.2–2.0 = large; >2.0 = very large [53]. Significance was accepted when $p$ value was <0.05.

## Results

The ICC demonstrated good to excellent reliability under all exercise conditions for the rectus femoris, biceps femoris, gluteus medius, vastus medialis, vastus lateralis, and rectus femoris of the rear leg (Table 1). The MVIC assessment demonstrated an excellent reliability for the rectus femoris (0.955; 95% CI: 0.90–0.98), rectus femoris of the rear leg (0.973; 95% CI: 0.94–0.98), vastus medialis (0.945; 95% CI: 0.88–0.97), vastus lateralis (0.956; 95% CI: 0.90–0.98), biceps femoris (0.956; 95% CI: 0.90–0.98), and gluteus medius (0.987; 95% CI: 0.97–0.99). The ICC for the forces exerted on the suspension straps for the suspended lunge (0.982; 95% CI: 0.95–0.99), suspended lunge-BOSU (0.956; 95% CI: 0.90–0.98), suspended lunge-Vibro30 (0.978; 95% CI: 0.95–0.99), and suspended lunge-Vibro40 (0.973; 95% CI: 0.94–0.98) demonstrated an excellent reliability. The ICC showed an excellent reliability for VGRF under the Bulgarian squat (0.996; 95% CI: 0.99–0.99) and suspended lunge (0. 995; 95% CI: 0.98–0.99).

**Table 1. Reliability values for each muscle analyzed under the Bulgarian squat and suspended lunge conditions.** Intra-rater reliability is expressed as ICC (95% CI).

| | Exercise condition | ICC (95% CI) |
|---|---|---|
| **Rectus Femoris** | Bulgarian squat | 0.943 (0.88–0.97) |
| | Suspended lunge | 0.882 (0.75–0.95) |
| | Suspended lunge-BOSU | 0.888 (0.76–0.95) |
| | Suspended lunge-Vibro30 | 0.899 (0.78–0.95) |
| | Suspended lunge-Vibro40 | 0.945 (0.88–0.97) |
| **Biceps Femoris** | Bulgarian squat | 0.919 (0.83–0.96) |
| | Suspended lunge | 0.871 (0.73–0.94) |
| | Suspended lunge-BOSU | 0.878 (0.74–0.94) |
| | Suspended lunge-Vibro30 | 0.795 (0.57–0.91) |
| | Suspended lunge-Vibro40 | 0.990 (0.97–0.99) |
| **Gluteus Medius** | Bulgarian squat | 0.895 (0.78–0.95) |
| | Suspended lunge | 0.894 (0.77–0.95) |
| | Suspended lunge-BOSU | 0.946 (0.88–0.97) |
| | Suspended lunge-Vibro30 | 0.941 (0.87–0.97) |
| | Suspended lunge-Vibro40 | 0.925 (0.84–0.96) |
| **Vastus Medialis** | Bulgarian squat | 0.947 (0.88–0.97) |
| | Suspended lunge | 0.914 (0.82–0.96) |
| | Suspended lunge-BOSU | 0.935 (0.86–0.97) |
| | Suspended lunge-Vibro30 | 0.904 (0.79–0.95) |
| | Suspended lunge-Vibro40 | 0.918 (0.82–0.96) |
| **Vastus Lateralis** | Bulgarian squat | 0.880 (0.74–0.94) |
| | Suspended lunge | 0.916 (0.82–0.96) |
| | Suspended lunge-BOSU | 0.926 (0.84–0.96) |
| | Suspended lunge-Vibro30 | 0.758 (0.49–0.89) |
| | Suspended lunge-Vibro40 | 0.922 (0.83–0.96) |
| **Rectus Femoris_RL** | Bulgarian squat | 0.887 (0.76–0.95) |
| | Suspended lunge | 0.855 (0.69–0.93) |
| | Suspended lunge-BOSU | 0.856 (0.70–0.93) |
| | Suspended lunge-Vibro30 | 0.911 (0.78–0.96) |
| | Suspended lunge-Vibro40 | 0.959 (0.91–0.98) |

RL = Rear leg; CI = Confidence interval

The main effects of exercise condition were identified for mean muscle activation of the rectus femoris [$F_{(2.57, 48.79)}$ = 8.557 $p$ = 0.000, $\eta_p^2$ = 0.31], biceps femoris [$F_{(4,76)}$ = 3.495 $p$ = 0.011, $\eta_p^2$ = 0.15], gluteus medius [$F_{(4,76)}$ = 17.467 $p$ = 0.000, $\eta_p^2$ = 0.47], vastus medialis [$F_{(4,76)}$ = 5.578 $p$ = 0.001, $\eta_p^2$ = 0.23], vastus lateralis [$F_{(4,76)}$ = 6.074 $p$ = 0.003, $\eta_p^2$ = 0.24], rectus femoris of the rear leg [$F_{(4,76)}$ = 5.501 $p$ = 0.001, $\eta_p^2$ = 0.23]; mean muscle activation of the front leg muscles (Global_FL) [$F_{(4,76)}$ = 18.611 $p$ = 0.000, $\eta_p^2$ = 0.49]; and mean muscle activation of all muscles (Global) [$F_{(4,76)}$ = 10.524 $p$ = 0.000, $\eta_p^2$ = 0.36]. The suspended lunge provided lower but non-significant activations than the Bulgarian squat for the biceps femoris ($p$ = 0.392, $d$ = 1.33), gluteus medius ($p$ = 1.000, $d$ = 0.27), vastus medialis ($p$ = 1.000, $d$ = 0.63), vastus lateralis ($p$ = 0.647, $d$ = 1.66), Global_FL ($p$ = 1.000, $d$ = 1.78), and Global ($p$ = 0.109, $d$ = 2.84). Furthermore, the suspended lunge showed significantly lower activations than the suspended lunge-BOSU, suspended lunge-Vibro30, and suspended lunge-Vibro40 in the muscles above (Table 2). Pairwise comparisons details between exercise conditions and all muscle

**Table 2. Normalized electromyographic activation for each lower body muscle under different lunge conditions as a percentage of maximum voluntary isometric contraction (%MVIC).** Values are expressed as mean ± standard error of the mean (SEM).

| | Bulgarian Squat (a) | Suspended Lunge (b) | Suspended Lunge-BOSU (c) | Suspended Lunge-Vibro30 (d) | Suspended Lunge-Vibro40 (e) | P-value (effect size *d*) | | | | | |
|---|---|---|---|---|---|---|---|---|---|---|---|
| | | | | | | a-c | b-c | d-c | d-e | | |
| RF_FL | 32.72 ± 3.48† | 33.50 ± 3.45† | 45.30 ± 4.28 | 35.16 ± 3.96†§ | 44.90 ± 5.72 | 0.010 (0.72) | 0.002 (0.68) | 0.001 (0.55) | 0.012 (0.44) | | |
| | | | | | | b-d | b-e | | | | |
| BF | 24.50 ± 2.40 | 21.48 ± 2.14†§ | 27.21 ± 2.21 | 28.07 ± 2.30 | 26.92 ± 2.38 | 0.044 (0.66) | 0.014 (0.54) | | | | |
| | | | | | | a-c | a-e | b-c | b-d | b-e | |
| Gmed | 46.53 ± 4.18†§ | 45.54 ± 3.15†‡§ | 65.67 ± 4.85 | 55.73 ± 4.67 | 65.59 ± 4.98 | 0.000 (0.95) | 0.001 (0.93) | 0.000 (1.10) | 0.022 (0.57) | 0.000 (1.08) | |
| | | | | | | a-e | b-e | | | | |
| VM | 64.58 ± 3.75§ | 62.18 ± 3.90§ | 67.61 ± 2.87 | 69.05 ± 4.45 | 76.23 ± 4.57 | 0.014 (0.62) | 0.006 (0.74) | | | | |
| | | | | | | b-d | b-e | | | | |
| VL | 72.34 ± 4.81 | 64.92 ± 4.13†§ | 76.79 ± 3.80 | 81.13 ± 6.31 | 87.63 ± 5.49 | 0.038 (0.68) | 0.03 (1.05) | | | | |
| | | | | | | c-a | | | | | |
| RF_RL | 33.51 ± 3.76 | 24.69 ± 3.87 | 23.61 ± 2.56* | 26.31 ± 3.09 | 28.60 ± 3.00 | 0.019 (0.69) | | | | | |
| | | | | | | a-c | a-e | b-c | b-d | b-e | d-e |
| GL_FL | 47.94 ± 1.40†§ | 45.52 ± 1.31†‡§ | 56.31 ± 1.96 | 53.83 ± 1.89§ | 60.26 ± 2.32 | 0.005 (1.10) | 0.000 (1.44) | 0.000 (1.44) | 0.001 (1.14) | 0.000 (1.75) | 0.043 (0.68) |
| | | | | | | a-e | b-c | b-d | b-e | | |
| GL | 46.75 ± 1.48§ | 42.76 ± 1.33†‡§ | 50.64 ± 2.20 | 50.53 ± 1.46 | 54.37 ± 2.03 | 0.010 (0.96) | 0.012 (0.97) | 0.001 (1.26) | 0.000 (1.51) | | |

RF_FL = Rectus femoris front leg; BF = Biceps femoris; Gmed = Gluteus medius; VM = Vastus medialis; VL = Vastus lateralis; RF_RL = Rectus femoris rear leg;

GL_FL = Global mean of the five front leg muscles; GL = Global mean of the six muscles

* = Significantly lower than Bulgarian squat;

† = Significantly lower than Suspension lunge-BOSU

‡ = Significantly lower than Suspension lunge-Vibro30;

§ = Significantly lower than Suspension lunge-Vibro40

activation data are presented in Table 2. The percentage of electromyographic activations for all suspended lunges related to the Bulgarian squat conditions is shown in Fig 2.

Fig 3 shows the forces exerted on the suspension straps by the rear leg for each suspended lunge condition and VGRF produced by the front leg in the Bulgarian and suspended lunge exercises. An exercise condition main effect was found for the forces exerted by the rear leg on the suspension strap [$F_{(3,57)}$ = 5.106 $p$ = 0.003, $\eta_p^2$ = 0.21]. The force exerted on the suspension strap was significantly lower during the suspended lunge-BOSU than during the suspended lunge-Vibro30 ($p$ = 0.009, $d$ = 0.56) (Fig 3a). Furthermore, the front leg force production was significantly higher during the suspended lunge than during the Bulgarian squat ($t_{(19)}$ = -3.106, $p$ = 0.006, $d$ = 0.48) (Fig 3b).

## Discussion

The main findings of the study were that the effect of the suspension strap does not provoke an increase of the muscle activity in the front leg in the suspended lunge and the lack of a

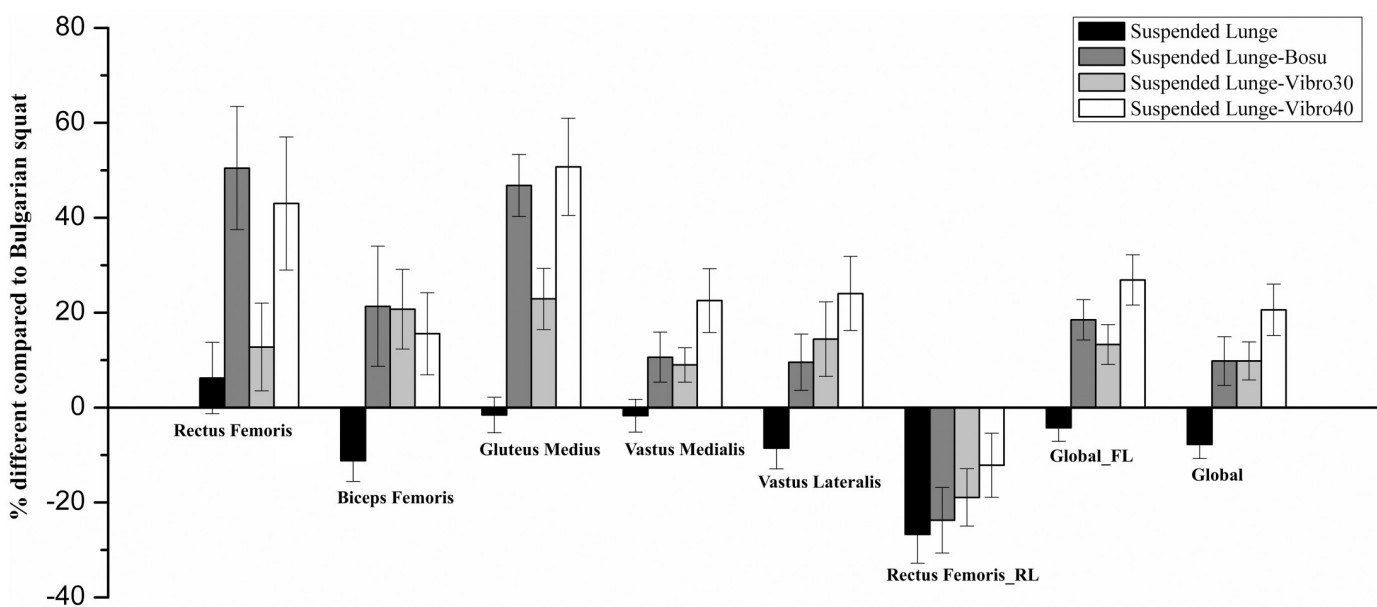

**Fig 2. Electromyographic activations for all conditions relative to the Bulgarian squat. Each bar represents the mean, and the error bar represents the standard error of the mean (SEM).** FL = Front leg; RL = Rear leg.

consistent support point was equally demanding for the analyzed muscles. Thus, similar muscle activation of suspended lunges as that of Bulgarian squats ranged from moderate (rectus femoris and biceps femoris) to high (gluteus medius) and very high (vastus medialis and lateralis), which reinforces this argument. All the suspended lunge conditions, except the suspended lunge-BOSU, showed a higher but non-significant activation of the rectus femoris compared to the Bulgarian squat. The suspended lunge-BOSU achieved a significantly higher activation of the rectus femoris compared to the moderate activity in the Bulgarian squat ($p = 0.010$, $d = 0.72$). The same recruitment patterns for the rectus femoris were found by Krause et al. [36] who reported non-significant differences in the activation of the rectus

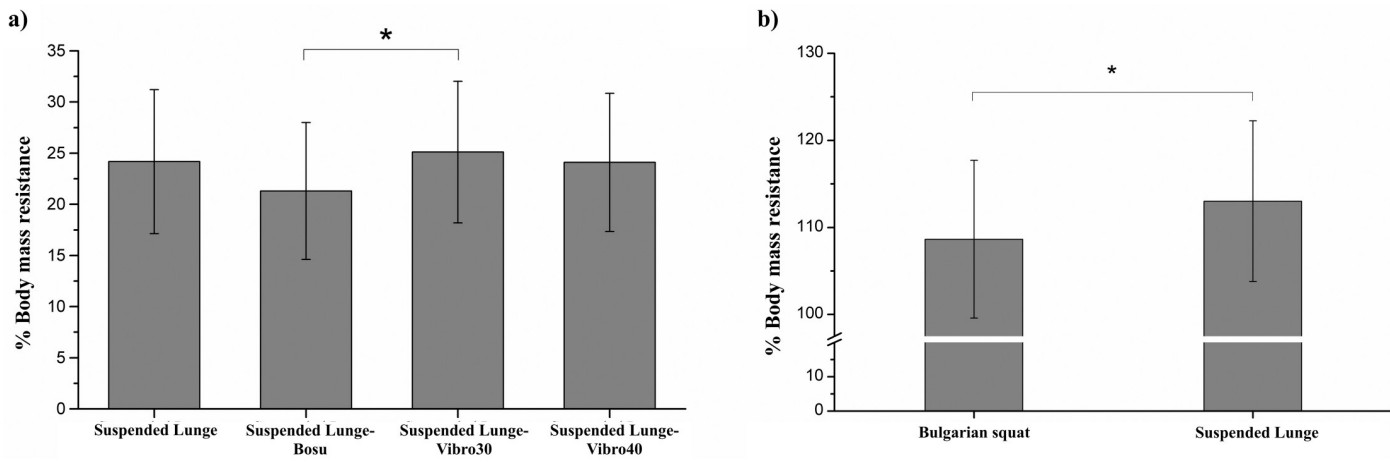

**Fig 3. Force values during the Bulgarian squat and suspended lunge conditions: a) Comparison between forces exerted by rear leg on the suspension strap and exercise condition, b) Front leg force production comparison between Bulgarian squat and suspended lunge. Each bar represents the mean, and the error bar represents the standard deviation (SD).** * Significant difference (p<0.05).

femoris in the standard lunge compared to that in the suspended lunge (22.1 ± 22.2%MVIC vs. 24.5 ± 22.0%MVIC, *p* = 0.434). Furthermore, Andersen et al. [17] did not find significant differences in the activation of the rectus femoris while performing a 6-RM Bulgarian squat under stable and unstable conditions (stable vs. unstable: 70.7 ± 18.3%MVIC vs. 68.9 ± 16.1% MVIC). On the other hand, it seems that performing a unilateral lower limb exercise with a suspension strap on the rear leg or an unstable surface on the front leg causes higher demands for the rectus femoris. This is because the primary role of the rectus femoris in the Bulgarian squat and suspended lunge could be the control of the hip flexion and knee extension movements, instead of stabilizing the abduction, adduction, and rotational movements of the hip and pelvis [36].

Regarding the remaining front leg muscles, the Bulgarian squat showed a slightly greater but non-significant muscle recruitment compared to the suspended lunge. For the biceps femoris, the activation was moderate; in the gluteus medius, the activation was high; and in the vastus medialis and vastus lateralis, the activation was very high among the conditions. As reported in previous studies, the vastus medialis and lateralis achieved a higher, but non-significant, very-high activation during a 6-RM Bulgarian squat compared to the unstable Bulgarian squat [17]. The study conducted by Mausehund et al. [54], in healthy and moderate strength-trained students, indicated that the activation of the vastus lateralis was higher, but not significant, for the 6-RM Bulgarian squat than for the 6-RM split squat and single-leg squat, even though both exercises registered a very high level of activity. These authors also showed non-significant differences for the gluteus medius while performing the Bulgarian squat and split squat, even though these two exercises provided a moderate activity of the gluteus medius. The Bulgarian squat was more gluteus medius demanding. Likewise, DeForest et al. [19] reported that during the concentric phase of a loaded Bulgarian squat, the activation of the biceps femoris (around 390 mV) and vastus medialis (around 640 mV) and lateralis (around 670 mV) was higher than that of a bilateral and split squat. In contrast, Krause et al. [36] reported that the suspended lunge increases significantly the muscle recruitment for the hamstring and gluteus medius (13.1 ± 20.1% MVIC; 24.1 ± 15.1%MVIC, respectively) compared to a standard lunge (hamstring: 8.7 ± 13.2%MVIC, *p* = 0.01; gluteus medius: 15.3 ± 11.4% MVIC, *p* = 0.01). Exercise technique may explain the differences in muscle activity because previous studies showed that when performing a standard lunge, in healthy subjects, the muscle activity of the biceps femoris was low [55,56], that of the gluteus medius ranged from low to moderate [18,55], and that of the vastus medialis and lateralis ranged from high to very high [55,56]. Differently, the Bulgarian squat is more demanding than the standard lunge. Previous studies showed that the activity of the biceps femoris and vastus (medialis and lateralis) was very high [17,54] and that of the gluteus medius was moderate [54]. Thus, performing a Bulgarian squat with the front leg on the floor demands a higher hip and thigh muscle recruitment than a standard lunge, and therefore, the difference in the muscle activation between the traditional and suspended exercises is higher in case of a standard lunge than the Bulgarian squat. Furthermore, leaning the rear leg on the suspension strap appears to produce a decrease in the recruitment of these muscles.

Another finding was the need for a dual condition to elicit higher muscle activation, in the front leg (suspended lunge-BOSU, suspended lunge-Vibro30, and suspended lunge-Vibro40) but not in the rear leg. The two conditions eliciting higher activation of the rectus femoris and gluteus medius in the front leg were suspended lunge-BOSU (45.30 ± 4.28%MVIC; 65.67 ± 4.85%MVIC, respectively) and suspended lunge-Vibro40 (44.90 ± 5.72%MVIC; 65.59 ± 4.98%MVIC, respectively). For these muscles, the stimulus provoked by the BOSU® conditions could be equivalent, in terms of muscle activation, with those offered by the WBV platform at 40 Hz-high, but not at 30 Hz-high. Pollock et al. [57] found in healthy participants

standing on a WBV platform at 30 Hz of frequency and 5.5 mm of amplitude that the rectus femoris recruitment was significantly higher than when WBV was set at 5 Hz of frequency and the same amplitude. These authors indicated that muscle recruitment for the rectus femoris depends on the frequency and amplitude of vibration. This finding suggests that dual conditions with WBV and compliant environments compromised the postural stability, leading to increased muscle tuning mechanisms and muscle contraction [29,58]. Furthermore, gluteus medius was solicited to stabilize the body during the dynamic flexo-extension of the front leg, which characterizes lunges under a suspended-BOSU condition, but also to absorb the vibration offered by the vibration plate. Moreover, the activation found in the antagonist (biceps femoris) and vastus (medialis and lateralis) was similar and not significantly different in the three dual conditions, being higher in the Vibro40 condition. The equivalences of the effects between BOSU® and vibratory conditions might be caused by the contribution of multiple neural pathways with distinct functional roles to rapid motor control response to a perturbation [59]. Thus, the neuromuscular response for maintaining the posture on a BOSU® may be more intelligent than merely a voluntary or a reflex mechanism [60] integrating the modulation of the long-latency stretch reflexes. Sensitivity increases of these reflexes were reported when subjects interacted with compliant environments, and this suggests its significant role in maintaining the limb stability in such conditions [59]. According to this, the reflex motor response during the BOSU® condition and the vibratory tonic reflex on the WBV platforms might induce similar activation in the involved muscles. This finding, also reflected in the global activation (the mean of all analyzed muscles), might be explained by the particular requirements of absorbing the vibration or maintaining the stability on a BOSU®. Hence, performing dynamic tasks on a BOSU®, subjects experiment a muscular trembling (micro amplitude changes), provoked by body mass variations projected on the forward leg, leaned on a compliant surface like this during the whole range of movement. These micro amplitude changes are described as one of the muscle tuning mechanisms for vibration training [20]. Additionally, WBV has been proven as beneficial improving the coordination of the synergistic muscles and increasing the inhibition of the antagonists, together with increases in hormonal responses of testosterone and growth hormone [61], besides the beneficial effects on bone mineral density [62], muscle blood volume [63] or balance control, and muscle endurance [64].

In terms of global activation, the use of WBV platforms, together with devices such as BOSU®, enhances muscle activity in the suspended lunge in physically active young adults. Thus, the simple use of a suspension device is not demanding enough for the studied exercise and needs to be complemented with other loading sources. So, inclusion of additional methods increasing the instability (BOSU®, Swiss ball, Pielaster®, rubber mats), vibration with demanding amplitudes and frequencies, and extra weights (weighted vests and belts, barbells, kettlebells) is necessary to increase the muscle activation of the involved muscles and the force produced.

The third finding of this study was that the force produced on the suspension straps was significantly lower for suspended lunge-BOSU than for suspended lunge-Vibro30 (21.3% ± 6.7 vs. 25.1% ± 6.93, $p = 0.009$), and this force was lower, but not significant, than the suspended lunge and suspended lunge-Vibro40. Thus, the present study shows that the percentage of body mass resistance exerted by the rear leg on the suspension strap could not be influenced by the front leg lean (on the floor or the WBV platform). However, to perform the suspended lunge under dual condition with a device such as BOSU® provokes an increase in the amount of instability, and thus, the load exerted by the rear leg on the suspension strap decreases in accordance with Behm et al. [40] and their hierarchy of force outputs proposal, which states that the degree of stability or instability affects limb force production directly. This finding is

according to Saeterbakken & Fimland [42] who reported that in healthy subjects, the isometric force output achieved while performing a squat on BOSU® (603 ± 208 N) was significantly lower than the force produced under a stable squat on the floor (749 ± 222 N) or less unstable surfaces as squats on the power board (694 ± 220 N).

The VGRF exerted by the front leg on the force plate was significantly higher during a suspended lunge than during the Bulgarian squat (113.01% ± 9.24 vs. 108.65% ± 9.05, $p$ = 0.006). This finding suggests that leaning the rear leg on a suspension strap provokes a transfer of a certain amount of body mass resistance towards the front leg, maintaining the trunk position, which exerts a force on the ground to attempt to keep the posture. Also, the increase of VGRF in the suspended lunge may be due to the low activation of rectus femoris of the rear leg. Consequently, maintaining the rear leg on a suspension device could inhibit the role of rectus femoris as a hip flexor and contribute to the increase of the VGRF in the front leg.

There were some limitations associated with this study. Results of the present study may be influenced by subjects' experiences with similar exercises to those performed in the present investigation. Each individual has a different level of motor control for the same task, and this might be taken into account when assessing muscle electrical signals. Therefore, participants' characteristics might constitute a limitation to infer the results of the present study. This study did not use functional tests to determine participants' laterality, together with their neuromuscular and performance level. Moreover, the lack of quantification about the amount of instability produced by the device should be considered. Another limitation may be that a goniometer did not control the knee flexion angle. However, the displacement during each repetition of the Bulgarian squat and suspended lunge conditions was measured with a positional encoder. Further research should examine the muscle activity and force output when performing suspended lunges to compare the muscle recruitment between lower body suspension and traditional resistance training exercises. Furthermore, the assessment of the perturbation related to the use of unstable surfaces with an accelerometer would be interesting.

In conclusion, the results of this study demonstrated that suspended lunges provide no additional benefit than Bulgarian squats to enhance lower body muscle activity. Performing a lunge at dual conditions increases exercise muscle activity compared with a Bulgarian squat and suspended lunge. However, dual conditions decrease the load on the suspension strap when the front leg leans on an unstable surface (i.e., BOSU®), and the VGRF exerted by the front leg in the suspended lunge (compared to its traditional counterparts) is enhanced to overcome the instability generated by the suspension device.

## Supporting information

**S1 File. STROBE checklist of the study.**
(DOCX)

**S2 File. Clinical studies checklist.**
(DOCX)

## Acknowledgments

We are grateful to all the study subjects for their participation.

## Author Contributions

**Conceptualization:** Joan Aguilera-Castells, Bernat Buscà, Azahara Fort-Vanmeerhaeghe, Javier Peña.

**Formal analysis:** Joan Aguilera-Castells, Bernat Buscà, Jose Morales, Fernando Rey-Abella, Javier Peña.

**Investigation:** Joan Aguilera-Castells, Bernat Buscà, Mònica Solana-Tramunt, Jaume Bantulà, Javier Peña.

**Methodology:** Joan Aguilera-Castells, Bernat Buscà, Jose Morales, Mònica Solana-Tramunt, Azahara Fort-Vanmeerhaeghe, Javier Peña.

**Project administration:** Joan Aguilera-Castells.

**Supervision:** Bernat Buscà, Javier Peña.

**Writing – original draft:** Joan Aguilera-Castells, Bernat Buscà, Fernando Rey-Abella, Javier Peña.

**Writing – review & editing:** Joan Aguilera-Castells, Bernat Buscà, Jaume Bantulà, Javier Peña.

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
