## [Decision Letter · Decision Letter 0]

23 Jul 2019

PONE-D-19-14691

Muscle activity of Bulgarian squat. Effects of additional vibration, suspension and unstable surface

PLOS ONE

Dear Dr. Buscà,

Thank you for submitting your manuscript to PLOS ONE. After careful consideration, we feel that it has merit but does not fully meet PLOS ONE’s publication criteria as it currently stands. Therefore, we invite you to submit a revised version of the manuscript that addresses the points raised during the review process.

We would appreciate receiving your revised manuscript by Sep 06 2019 11:59PM. To enhance the reproducibility of your results, we recommend that if applicable you deposit your laboratory protocols in protocols.io, where a protocol can be assigned its own identifier (DOI) such that it can be cited independently in the future. For instructions see: http://journals.plos.org/plosone/s/submission-guidelines#loc-laboratory-protocols

We look forward to receiving your revised manuscript.

Kind regards,

Carlos Balsalobre-Fernández

Academic Editor

PLOS ONE

Reviewers' comments:

Reviewer's Responses to Questions

**Comments to the Author**

1. Is the manuscript technically sound, and do the data support the conclusions?

Reviewer #1: Yes

Reviewer #2: Partly

2. Has the statistical analysis been performed appropriately and rigorously? 

Reviewer #1: N/A

Reviewer #2: Yes

3. Have the authors made all data underlying the findings in their manuscript fully available?

Reviewer #1: Yes

Reviewer #2: Yes

4. Is the manuscript presented in an intelligible fashion and written in standard English?

Reviewer #1: No

Reviewer #2: Yes

5. Review Comments to the Author

Reviewer #1: General considerations

- Consistence BOSU ®

- There are very long sentences without full stops or commas.

- I would recommend to improve the wording, fluency (use connectors) and to be careful with the sections that include many acronyms, which difficult the understanding of the ideas (establish an order, make the method very clear, comparisons).

- An interesting article by Moras G. has been recently published.

- We suggest that it would more explicit in the bibliographical citations (results, type of sample, sport, more information).

Abtsract

Line 26-28: some pause in the sentence

Line 28: ‘different suspended lunges’. Clarify the meaning.

Line 30: Bosu (like in a methods section) more consistent. ‘®’

Line 30: you didn’t define Vibro30 previously.

Line 31: males

Line 32: 24.40 _ space ‘±’

Line 33: consistency , you don’t define the BOSU like an unstable Surface , you worte directly Vibro30, and then just electromiography, can you be more consistent.

Line 34: five’

Introduction

We suggest to introduce a first part of functional exercises, differences, bilateral/ unilateral relationship, relations with performance / health (the area which you want to focus the proposal on), detail and justify exercises you’ve chosen. Shorten other sections. (line 57-75)…

and introduce in separate sections the different methods

- Inestability

- Vibration

- suspension

Line 48: Can you report some ref./ demands on?

Line 49-51: can you clarify this sentence

Line 51-53: some commas/full stops in the sentence

Line 53: can you report more ref, and more information…

Line 55: ref2 . in relation to upper body?

Line 56: ref 4, can you cite some more info (sample, gender, athlets…)

Line 56:

- you could connect this phrase and make it clearear.

- 'promising markers' you can use different terminology than the original article

- can you report the tests and the sample T-Test, sprint 10m….

Line 57: Different authors disagree wiht the comparisons in different levels of participants (Maloney et al).. we consider that it should introduce a sentence, in relation to which the level of the participants can become a limitation in the results ... methods .....

Line 57-63. All the paragraph are related to the ref. 6?

Line 59. Can you support this idea with references ..(Comfort, P……

Line 63. Before ‘Perfoming’, can you introduce some connectors in this sentence.

Line 66: order and calrify the sentence

Line 63-65: can you cite more information (methods to analyse these activation, all the muscles involved in concentric/eccentric phase in the same %, and in the same function)

Line 69: Little ¿ can you improve this

Line 70: in bulgarian squat?

Line 69-72: can you rewrite and try to make it more fluent.

Line 73: hamstrings? Can you be more precisley

Line 76: can you report the diferences why (this transferability in sport line 58)) , for example differences in unilateral and bilateral; vectors force; can you report examples, relations with a … performance, (health,techinque)

Line 85: exercises? Can you report more information

Line 100-104: Can you write in a clear and concrete way.

Line 118-119: can you justify why you add this ref, and can you report more data in relation these studies (in upper body)

Line 123: you explain the effects in muscle activation, and to ‘decreased force output’, but I recommended to explain the positive effects in neuromuscular control?

Line 128, can you cite some ref?

Methods

Line 151-154: some pause in the sentence, and repport more information and clarify the idea

Line 164: could be influnced by the difference of age 20-31 in the results , could be influenced by the experience , sport, in the results?

Line 168: can you repport more information

Procedures

Line 184: could you explain the familiarisation procedures

Line 189: ref? dominance?

if it’s posible can you disscuss (not in the text) why this test and not another one.

The analyzed capacity (exercise) could influence the dominance (Bishop, C, and another recent authors ...)

We believe that dominance could have been determined with the same task previously. This could be a limitation of the study.

Excuse me, but I haven’t been able to find which legs are being analyzed (we understand that the dominant leg) it could improve the explanation of the leg analyzed in the text..and in different sections.

If you have only analyzed the dominant leg, do not consider analyzing the non-dominant leg in the future and establish the differences, asymmetries, and the behavior of dom / non-dom legs in different situations?

Line 204-208 can you rewrite

Line 357: Why don’t use the newer scale?

(Cohen d < 0.2 = trivial; 0.2-0.6 = small; 0.6-1.2 = moderate; 1.2-2.0 = large; > 2.0 = very large)

Discussion

I recommend adding the scales or results of the differences found, as well as the results of the studies you are comparing, to see the comparison.

We also recommend to add more information from the comparative studies, in relation to the sample, sports

Line 438: than 'the' bulgarian

Line 174: the vibration does not provide other benefits? you could add some information that explains the benefits in other 'systems'

Line 477-493: the same idea is repeated on several occasions, could you clarify it and bemore precise. This part should to be considerably shorter.

Line 505-508: can rewrite

Limitations

In the introduction, note the importance of adaptations to a specific level. Do you consider that a limitation might not apply functional tests?

Reviewer #2: Summary of the research

This research study aims to compare the muscle activity and force production of different exercise conditions (bulgarian squat, suspended lunge, suspended lunge with instability and suspended lunge with two different vibration frequencies). The topic is interesting; however the study has significant flaws and requires major and minor revisions in order to meet the journal’s requirements.

Major issues

Abstract section.

First of all, while the study is interesting, the structure of the abstract is unclear, making it difficult to follow. I advise the authors to re-write the abstract section in order to clarify primary and secondary aims of the study and how they were approached to improve the flow and readability of the text. At the last paragraph of the introduction section appears to be clearer than in the abstract section.

Secondly, when authors mention differences found, it would be helpful to report the magnitude of this differences and the exact p values in order to allow the reader to evaluate clinical significance of the results.

Minor issues

Introduction section.

Lines 56 and 57.

The authors stated “Unstable surfaces have also been used to strengthen the lower body and have been demonstrated as promising markers of athletic performance in sprint and agility”, however the reference used (Cressey 2007), would not be appropriate. While results in sprint and T-test improved as cited, these improvements were always much lower than the stable group and the other variables assessed obtained impaired results. In fact, the author himself concludes “These results indicate that unstable training using inflatable rubber discs attenuates performance improvements in healthy, trained athletes. Such implements have proved valuable in rehabilitation, but caution should be exercised when applying unstable training to athletic performance and general exercise scenarios”. I would suggest to use another reference to justify their argument.

Lines 82-84

The authors stated: “…and the beneficial effects of WBV have been demonstrated in lower limb exercises (squat, half squat, Bulgarian squat or lunge) (15,16)”, however the vast majority of the participants in this references were untrained or post-menopausal making it difficult to extrapolate their results to other populations, this fact should be annotated.

Procedures section

Line 188

The authors determined the leg dominance of the participants but nothing else is explained about how this characteristic is used at the protocol of the study.

Line 207

The words “complementary methods” should be corrected by “complementary means of training”. Suspension, unstable surface and WBV are instruments used in order to apply a training load not a systematic approach to overload the athlete.

Exercise trials section

In order not to be confusing, it should be annotated that the participants executed the exercise only with their own bodyweight as a load.

Related to this, the fact of using the same absolute intensity in all exercises (bodyweight), implies that different relative intensities were used at the different conditions of the study. The most convenient way in order to compare the effects of instability, suspension and vibration conditions would have been to use the same relative load, at least in terms of load lifted, in all exercises. This is a limitation to be annotated.

Results section

Lines 383 y 384

The authors stated: “The suspended lunge provided the lowest activations for BF, Gmed, VM, VL, Global_FL and Global among the conditions”. The sentence is correct at descriptive level, but pairwise comparisons’ p values of table 2 does not reflect the same and it has not been mentioned or analysed in text either. For example, none of the EMG values of the suspended lunge condition achieve significant differences respect to Bulgarian squat.

Discusion section

Line 420

The authors stated: “The lower muscle activation of suspended lunges compared to Bulgarian squats for the RF_RL, BF, Gmed, VM, and VL (but not the RF_FL) reinforces this argument”. But any significant analysis is reported and % difference of Gmed and VM shown in fig 2 seems not statistically significant.

Line 422

The authors stated: “All suspended lunge conditions increase RF_FL activity in comparison with the Bulgarian squat”. But any significant analysis is reported and % difference shown in fig. 2 seems not significant.

Line 433

The authors stated: "Regarding the rest of the FL muscles, BF, VM, VL, and Gmed showed a greater activity under Bulgarian squat condition, as reported in previous studies (8–10)”. In table 2 seems to be non-statistically significant.

6. PLOS authors have the option to publish the peer review history of their article (what does this mean?). If published, this will include your full peer review and any attached files.

Reviewer #1: No

Reviewer #2: No

---

## [Author Response · Author response to Decision Letter 0]

9 Aug 2019

Responses to Reviewer

Please note that: 

[R1] = comments from Reviewer #1.

[R2] = comments from Reviewer #2.

[A] = answers from the authors.

{…} = text modified in the revised manuscript.

Reviewer #1: General Considerations

[R1] “- Consistence BOSU®”

[A]

The authors use the term BOSU® to refer to the unstable surface. However, we maintain the term suspended lunge-BOSU to refer to the exercise condition (independent variable). 

[R1] “- There are very long sentences without full stops or commas.”

[A]

Some sentences have been shortened.

[R1] “- I would recommend to improve the wording, fluency (use connectors) and to be careful with the sections that include many acronyms, which difficult the understanding of the ideas (establish an order, make the method very clear, comparisons).”

[A]

The wording fluency has been improved. We have introduced more connectors to improve the reading experience. Moreover, the muscles abbreviation have been replaced by the muscle names; also the abbreviations to refer to front leg (FL) and the rear leg (RL) have been replaced into the body of the manuscript. We only maintain the abbreviations of the vertical ground reaction forces (VGRF), the whole body vibration (WBV), the percentage of maximum voluntary isometric contraction (%MVIC), the interclass correlation coefficient (ICC) and the surface electromyography (sEMG).

[R1] “- An interesting article by Moras G. has been recently published.”

[A]

The study carried out by Moras et al. (2019) has been added in the introduction to add more information on the combination of training methods. Thank you for your suggestion. 

[R1] “- We suggest that it would more explicit in the bibliographical citations (results, type of sample, sport, more information).”

[A]

Thank for your comment. We have try more explicit in the bibliographical citation reporting the type of the sample, sport and results. 

Specific comments:

Abstract

[R1] “Line 26-28: some pause in the sentence”

[A]

Amended.

[R1] “Line 28: ‘different suspended lunges’. Clarify the meaning.”

[A]

The abstract has been re-written and the meaning of the different suspended lunges has been clarified.

[R1] “Line 30: Bosu (like in a methods section) more consistent. ‘®’ ”

[A]

Amended.

[R1] “Line 30: you didn’t define Vibro30 previously.”

[A]

The meaning of suspended-Vibro30 and suspended-Vibro40 has been added to the abstract.

[R1] “Line 31: males”

[A]

Amended.

[R1] “Line 32: 24.40 _ space ‘±’ ”

[A]

Amended.

[R1] “Line 33: consistency, you don’t define the BOSU like an unstable Surface, you wrote directly Vibro30, and then just electromyography, can you be more consistent.”

[A]

Some parts of the abstract have been re-written to improve the consistency and we have defined BOSU® as an unstable surface, providing the meaning of Vibro30 and Vibro40.

[R1] “Line 34: five’ ”

[A]

Amended.

Introduction

[R1] “We suggest to introduce a first part of functional exercises, differences, bilateral/ unilateral relationship, relations with performance / health (the area which you want to focus the proposal on), detail and justify exercises you’ve chosen. Shorten other sections. (line 57-75)…

and introduce in separate sections the different methods

- Inestability

- Vibration

- suspension”

[A]

We have introduced a paragraph as suggested. The different training methods have been separate in sections. 

[R1] “Line 48: Can you report some ref./ demands on?”

[A]

We have added the reference of the study conducted by Behm et al. (2015). 

[R1] “Line 49-51: can you clarify this sentence”

[A]

The sentence has been re-written to provide a better meaning. 

{… Thus, the use of devices that create instability has become popular …}

[R1] “Line 51-53: some commas/full stops in the sentence”

[A]

The sentence has been modified as follows: 

{… Primarily, unstable devices are used to increase the load of traditional exercises by providing higher muscular demands through superior motor unit recruitment. Such devices also improve neuromuscular coordination to maintain balance during training exercises …}

[R1] “Line 53: can you report more ref, and more information…”

[A]

We have added more information and some references related to use of unstable surfaces and the neuromuscular adaptations. 

[R1] “Line 55: ref2 . in relation to upper body?”

[A]

We have provided more information and details about the reference (ref2). 

Anderson et al. analyzed trunk muscles and upper body muscles (triceps brachii).

[R1] “Line 56: ref 4, can you cite some more info (sample, gender, athlets…)”

[A]

This reference has been removed. Saeterbakken et al. (2019) examined the hip and thigh muscles under three squat conditions (stable and unstable conditions) so, these muscles are not from the upper body nor the trunk. Sorry for this mistake. 

[R1] “Line 56:

- you could connect this phrase and make it clearear.

[A]

The sentence has been re-written to make it clearer and a connector has been added. 

- 'promising markers' you can use different terminology than the original article

- can you report the tests and the sample T-Test, sprint 10m….”

[A]

This sentence and the reference have been removed. Reviewer #2 has suggested that we remove this reference because it was not accurate enough. 

[R1] “Line 57: Different authors disagree wiht the comparisons in different levels of participants (Maloney et al).. we consider that it should introduce a sentence, in relation to which the level of the participants can become a limitation in the results ... methods .....”

[A]

In the paragraph where we indicate the limitations of the study we have include your suggestion.

[R1] “Line 57-63. All the paragraph are related to the ref. 6?”

[A]

The paragraph has changed according to your suggestions. The reference was not precise so we have changed the wording of the text.

[R1] “Line 59. Can you support this idea with references ..(Comfort, P……”

[A]

This sentence has been removed and the idea has been developed in the first paragraph using other references. 

[R1] “Line 63. Before ‘Perfoming’, can you introduce some connectors in this sentence.”

[A]

A connector has been added.

[R1] “Line 66: order and calrify the sentence”

[A]

The sentence has been re-written in order to clarify the meaning. 

[R1] “Line 63-65: can you cite more information (methods to analyse these activation, all the muscles involved in concentric/eccentric phase in the same %, and in the same function)”

[A]

We have provided more information about the involvement of different muscles in a standard lunge and a Bulgarian squat.

[R1] “Line 69: Little ¿ can you improve this”

[A]

The sentence has been re-written. 

[R1] “Line 70: in bulgarian squat?”

[A]

We have clarified this sentence.

[R1] “Line 69-72: can you rewrite and try to make it more fluent.”

[A]

The paragraph has been changed according to the reviewer suggestion.

[R1] “Line 73: hamstrings? Can you be more precisely”

[A]

Yes, the authors of the cited study (Youdas et al. 2007) did not provide the exact hamstring muscles analyzed. 

[R1] “Line 76: can you report the diferences why (this transferability in sport line 58)) , for example differences in unilateral and bilateral; vectors force; can you report examples, relations with a … performance, (health, techinque)”

[A]

The transferability of the different type of exercises to sports considering laterality, directions, techniques and its relation to functional performance has been developed previously as suggested by the reviewer.

[R1] “Line 85: exercises? Can you report more information”

[A]

We have reported more information about the exercises examined in the cited references. 

[R1] “Line 100-104: Can you write in a clear and concrete way.”

[A]

These sentences have been re-written to improve the meaning and we have provided more details and information about the cited study.

[R1] “Line 118-119: can you justify why you add this ref, and can you report more data in relation these studies (in upper body)”

[A]

The effect of suspension devices on force production has been examined in upper body exercises like push-ups and inverted row in the past. For this reason, we cited these references. Although we studied a lower body exercise (Bulgarian squat) we honestly think that they have a similar scope. These references provide evidence of the methods used to measure force under suspended conditions. Besides, these cited references used a force plate and a load cell to assess VGRF and the force on the suspension strap, respectively. These references could be helpful to understand the force behavior under suspended conditions.

We have added more data about Melrose and Dawes (2015) and Gulmez (2017).

[R1] “Line 123: you explain the effects in muscle activation, and to ‘decreased force output’, but I recommended to explain the positive effects in neuromuscular control?”

[A]

We appreciate the reviewer comment, but the present study aimed at examining the effects of suspension, unstable surfaces and vibration on muscle activity and force (VGRF and force exerted on the suspension strap), for this reason, we have provided a consistent explanation of the effects in muscle activation and force. Perhaps, adding additional information about the positive effects in neuromuscular control would be out of our main aims. The effects in neuromuscular control could be interesting for future research related to the effects of unstable surfaces and suspended devices on rehabilitation, return to play… 

[R1] “Line 128, can you cite some ref? ”

[A]

Additional literature is cited in the previous lines. Saeterbakken and Fimland (2013), Chulvi-Medrano et al. (2010). 

Methods

[R1] “Line 151-154: some pause in the sentence, and repport more information and clarify the idea”

[A]

We have shortened the sentences and we have re-written them to clarify the idea. 

[R1] “Line 164: could be influnced by the difference of age 20-31 in the results , could be influenced by the experience , sport, in the results? ”

[A]

Only one subject was older than 30. The rest of the participants were between 22 and 26 years old. In terms of sports and experience, the sample was less homogenous, but no subject was a high-level athlete.

[R1] “Line 168: can you report more information”

[A]

Sport practiced by the study participants has been added.

Procedures

[R1] “Line 184: could you explain the familiarisation procedures”

[A]

We have added the familiarization procedure. 

[R1] “Line 189: ref? dominance?

[A]

The reference Meylan et al. (2009) has been introduced. 

[R1] if it’s posible can you disscuss (not in the text) why this test and not another one.

The analyzed capacity (exercise) could influence the dominance (Bishop, C, and another recent authors ...)

We believe that dominance could have been determined with the same task previously. This could be a limitation of the study.”

[A]

We have chosen the kicking leg to determine the dominant leg because this criterion is used in many previous studies such as Mayer et al. (2009), Miyaguchi and Demura (2010) or Theoharopoulos et al. (2000). Regarding the performance of lower limb exercises, Kibele et al. (2009) determined the dominant leg as the leg preferred to kick a ball, in order to analyze squat performance. Furthermore, the studies conducted by Youdas et al. (2007) and Boudreau et al. (2009) when testing muscle activity in the standard lunge, determined the front leg as the leg preferred to kick a ball. For this reason, we decided on this test.

According to the reviewer's suggestions, we have added this element in the limitations section. Thanks.

[R1] “Excuse me, but I haven’t been able to find which legs are being analyzed (we understand that the dominant leg) it could improve the explanation of the leg analyzed in the text..and in different sections.

[A]

We have improved the explanation of the analyzed legs in different sections of the manuscript.

If you have only analyzed the dominant leg, do not consider analyzing the non-dominant leg in the future and establish the differences, asymmetries, and the behavior of dom / non-dom legs in different situations? ”

[A]

We appreciate the reviewer comment. However, the aim of the present study was not analyzing muscular asymmetries. Leg dominance has only been used to refer to the front leg (dominant leg) to measure the muscle activity in the dominant leg muscles. Although, the reviewer suggestion can be taken into account for future research. 

[R1] “Line 204-208 can you rewrite”

[A]

The sentences have been re-written. 

[R1] “Line 357: Why don’t use the newer scale?

(Cohen d < 0.2 = trivial; 0.2-0.6 = small; 0.6-1.2 = moderate; 1.2-2.0 = large; > 2.0 = very large)”

[A]

According to the reviewer suggestion, we have added the newer scale to interpret the magnitude of the effect size based on Hopkins et al. (2009). 

Discussion

[R1] “I recommend adding the scales or results of the differences found, as well as the results of the studies you are comparing, to see the comparison.

We also recommend to add more information from the comparative studies, in relation to the sample, sports”

[A]

The discussion section is now incorporating the scales of muscle activity (low, moderate, high, and very high) and the results of the differences (p-values or effects size). Additionally, we have added the results of the studies used to compare and discuss our results. Besides, we have shown information about the sample, as the reviewer suggested. However, many of the cited studies recruited healthy subjects but did not inform about their sports background. 

[R1] “Line 438: than 'the' bulgarian”

[A]

Amended.

[R1] “Line 474: the vibration does not provide other benefits? you could add some information that explains the benefits in other 'systems' ”

[A]

Some benefits of WBV have been added. 

[R1] “Line 477-493: the same idea is repeated on several occasions, could you clarify it and be more precise. This part should to be considerably shorter.”

[A]

This paragraph has been re-written and shortened in order to clarify what we exactly meant.

[R1] “Line 505-508: can rewrite”

[A]

These sentences have been re-written.

Limitations

[R1] “In the introduction, note the importance of adaptations to a specific level. Do you consider that a limitation might not apply functional tests? ”

[A] 

Although the purpose of the present study was to study the suspended lunge and the Bulgarian squat, this might be a limitation of our study. Therefore, we have added this element as a limitation.

Reviewer #2: Specific comments

Abstract section

[R2] “First of all, while the study is interesting, the structure of the abstract is unclear, making it difficult to follow. I advise the authors to re-write the abstract section in order to clarify primary and secondary aims of the study and how they were approached to improve the flow and readability of the text. At the last paragraph of the introduction section appears to be clearer than in the abstract section.

Secondly, when authors mention differences found, it would be helpful to report the magnitude of this differences and the exact p values in order to allow the reader to evaluate clinical significance of the results.”

[A]

The authors have re-written the major part of the abstract according to the reviewer comments. 

The magnitude of the differences and the exact p-values have been added to the abstract. 

Introduction section

[R2] “Lines 56 and 57.

The authors stated “Unstable surfaces have also been used to strengthen the lower body and have been demonstrated as promising markers of athletic performance in sprint and agility”, however the reference used (Cressey 2007), would not be appropriate. While results in sprint and T-test improved as cited, these improvements were always much lower than the stable group and the other variables assessed obtained impaired results. In fact, the author himself concludes “These results indicate that unstable training using inflatable rubber discs attenuates performance improvements in healthy, trained athletes. Such implements have proved valuable in rehabilitation, but caution should be exercised when applying unstable training to athletic performance and general exercise scenarios”. I would suggest to use another reference to justify their argument.”

[A]

We have removed this reference following the reviewer suggestions. Also, we have re-written some paragraphs of the introduction, and we have removed this idea because it does not suit our intentions. 

[R2] “Lines 82-84

The authors stated: “…and the beneficial effects of WBV have been demonstrated in lower limb exercises (squat, half squat, Bulgarian squat or lunge) (15,16)”, however the vast majority of the participants in this references were untrained or post-menopausal making it difficult to extrapolate their results to other populations, this fact should be annotated.”

[A]

We have added the participants included in the cited studies. However, we would make the reviewer note that the systematic review conducted by Rehm et al. (2006) includes recreationally resistance-trained men (Ronestad et al., 2004), sprint trained athletes (Delecluse et al., 2005) and the rest of the studies included healthy untrained people and older adults. Furthermore, the meta-analysis carried out by Osawa et al. (2013) analyzes the WBV effects on the lower limb in young and older adults separately. For the WBV effects in youth, studies included untrained, physically or recreationally active participants and skiers.

Procedures section

[R2] “Line 188

The authors determined the leg dominance of the participants but nothing else is explained about how this characteristic is used at the protocol of the study.”

[A]

We have added this information according to the reviewer comment. 

[R2] “Line 207

The words “complementary methods” should be corrected by “complementary means of training”. Suspension, unstable surface and WBV are instruments used in order to apply a training load not a systematic approach to overload the athlete.”

[A]

The paragraph has been re-written to clarify its meaning. “Complementary methods” has been removed, and we have used different wording to express this idea. 

Exercise trials section

[R2] “In order not to be confusing, it should be annotated that the participants executed the exercise only with their own bodyweight as a load.

Related to this, the fact of using the same absolute intensity in all exercises (bodyweight), implies that different relative intensities were used at the different conditions of the study. The most convenient way in order to compare the effects of instability, suspension and vibration conditions would have been to use the same relative load, at least in terms of load lifted, in all exercises. This is a limitation to be annotated.”

[A]

We have added this information to clarify the exercise load. Thank you.

Sorry, but we do not understand the second comment. From our point of view, all subjects performed all conditions under their relative bodyweight load. This load did not change, and neither the pace nor the ROM. These procedures are the ones used in all the cited studies when comparing different conditions without extra loads; for instance, the studies conducted by Snarr et al. (2016), Calatayud et al. (2014) or Borreani et al. (2015).

Snarr, R., Hallmark, A. V., Nickerson, B. S., & Esco, M. R. (2016). Electromyographical Comparison of Pike Variations Performed With and Without Instability Devices. Journal of Strength and Conditioning Research, 30(12), 3436–3442. doi:10.1519/JSC.0000000000001436

Calatayud, J., Borreani, S., Colado, J., Martin, F., Rogers, M., Behm, D., & Andersen, L. (2014). Muscle activation during push-ups with different suspension training systems. Journal of Sports Science and Medicine, 13(3), 502–510.

Borreani, S., Calatayud, J., Colado, J. C., Moya-Nájera, D., Triplett, N. T., & Martin, F. (2015). Muscle activation during push-ups performed under stable and unstable conditions. Journal of Exercise Science & Fitness, 13(2), 94–98. doi: 10.1016/j.jesf.2015.07.002

Results section

[R2] “Lines 383 y 384

The authors stated: “The suspended lunge provided the lowest activations for BF, Gmed, VM, VL, Global_FL and Global among the conditions”. The sentence is correct at descriptive level, but pairwise comparisons’ p values of table 2 does not reflect the same and it has not been mentioned or analysed in text either. For example, none of the EMG values of the suspended lunge condition achieve significant differences respect to Bulgarian squat.”

[A]

According to the reviewer suggestion, we have re-written the explanation of these results. We have provided the p-values and the effect size between the Bulgarian squat and the suspended lunge for the mentioned muscles. 

We have wanted to highlight that the suspended lunge showed the lower but no significant activations compared to the Bulgarian squat.

Discussion section

[R2] “Line 420

The authors stated: “The lower muscle activation of suspended lunges compared to Bulgarian squats for the RF_RL, BF, Gmed, VM, and VL (but not the RF_FL) reinforces this argument”. But any significant analysis is reported and % difference of Gmed and VM shown in fig 2 seems not statistically significant.”

[A]

We have amended this idea to be consistent with the results obtained in the present study. Thanks for the comment.

[R2] “Line 422

The authors stated: “All suspended lunge conditions increase RF_FL activity in comparison with the Bulgarian squat”. But any significant analysis is reported and % difference shown in fig. 2 seems not significant.”

[A]

We have indicated that the different suspended lunge conditions showed higher but non-significant activations for the rectus femoris compared to the Bulgarian squat. Although, the suspended lunge-BOSU reached a significant greater rectus femoris activation than the Bulgarian squat (p = 0.010, d = 0.72).

[R2] “Line 433

The authors stated: "Regarding the rest of the FL muscles, BF, VM, VL, and Gmed showed a greater activity under Bulgarian squat condition, as reported in previous studies (8–10)”. In table 2 seems to be non-statistically significant.”

[A]

For the Bulgarian squat, muscle activation of the mentioned muscles was higher, but not high enough to be significant, compared to the suspended lunge. So, we have modified the wording.

---

## [Editor Report · Decision Letter 1]

14 Aug 2019

Muscle activity of Bulgarian squat. Effects of additional vibration, suspension and unstable surface

PONE-D-19-14691R1

Dear Dr. Buscà,

We are pleased to inform you that your manuscript has been judged scientifically suitable for publication and will be formally accepted for publication once it complies with all outstanding technical requirements.

With kind regards,

Carlos Balsalobre-Fernández

Academic Editor

PLOS ONE
---

## [Editor Report · Acceptance letter]

19 Aug 2019

PONE-D-19-14691R1 

Muscle activity of Bulgarian squat. Effects of additional vibration, suspension and unstable surface 

Dear Dr. Buscà:

I am pleased to inform you that your manuscript has been deemed suitable for publication in PLOS ONE. Congratulations! Your manuscript is now with our production department. 

With kind regards,

on behalf of

Dr. Carlos Balsalobre-Fernández 

Academic Editor

PLOS ONE